# Endophytic Fungi: Key Insights, Emerging Prospects, and Challenges in Natural Product Drug Discovery

**DOI:** 10.3390/microorganisms10020360

**Published:** 2022-02-04

**Authors:** Pragya Tiwari, Hanhong Bae

**Affiliations:** Molecular Metabolic Engineering Lab, Department of Biotechnology, Yeungnam University, Gyeongsan 38541, Korea; pragyamita02@gmail.com

**Keywords:** anti-infectives, bioactive metabolites, biosynthetic gene clusters (BCG), drug discovery, endophytic fungi, fungal pharmacology, production platforms

## Abstract

Plant-associated endophytes define an important symbiotic association in nature and are established bio-reservoirs of plant-derived natural products. Endophytes colonize the internal tissues of a plant without causing any disease symptoms or apparent changes. Recently, there has been a growing interest in endophytes because of their beneficial effects on the production of novel metabolites of pharmacological significance. Studies have highlighted the socio-economic implications of endophytic fungi in agriculture, medicine, and the environment, with considerable success. Endophytic fungi-mediated biosynthesis of well-known metabolites includes taxol from *Taxomyces andreanae*, azadirachtin A and B from *Eupenicillium parvum*, vincristine from *Fusarium oxysporum*, and quinine from *Phomopsis* sp. The discovery of the billion-dollar anticancer drug taxol was a landmark in endophyte biology/research and established new paradigms for the metabolic potential of plant-associated endophytes. In addition, endophytic fungi have emerged as potential prolific producers of antimicrobials, antiseptics, and antibiotics of plant origin. Although extensively studied as a “production platform” of novel pharmacological metabolites, the molecular mechanisms of plant–endophyte dynamics remain less understood/explored for their efficient utilization in drug discovery. The emerging trends in endophytic fungi-mediated biosynthesis of novel bioactive metabolites, success stories of key pharmacological metabolites, strategies to overcome the existing challenges in endophyte biology, and future direction in endophytic fungi-based drug discovery forms the underlying theme of this article.

## 1. Introduction

Endophytes represent biological reservoirs of novel natural products, opening new avenues in the frontiers of drug discovery. Plant-associated microorganisms that colonize the internal tissues of all plant species are gaining momentum as key targets for bio-prospection in the search for novel chemical entities [1,2]. The discovery of the acclaimed anticancer drug paclitaxel accelerated research on endophytic biology and yielded prospective “drug candidates” with antimicrobial, immunosuppressant, antioxidant, and anti-neurodegenerative functions [1,2,3]. Currently, the impact of natural products in clinical applications cannot be underestimated [4], with the pharmaceutical sector adopting high-throughput approaches to screen plant secondary metabolites as potential lead molecules. According to a study by Newman and Cragg [5], among the 1562 drugs approved by the FDA, 141 comprised botanical drugs, 64,320 drugs were derivatives of natural products, while 61 drugs were synthesized based on natural pharmacophore [5]. 

With the emerging threat of drug-resistant microbes and antimicrobial resistance (AMR), it has become essential to discover novel antimicrobials to counter AMR. However, the low pace of discovery and indiscriminate use of the existing antibiotics have further necessitated the exploration of novel antimicrobial entities to compensate for the drying drug pipeline [2,6,7,8]. The rich yet less-explored diversity of endophytic entities and their considerable potential to impact the pharmaceutical industry have facilitated the discovery of secondary metabolites of therapeutic significance (Figure 1. Bio-prospection of endophytes and discovery of novel, high-value metabolites of commercial significance, including hinnuliquinone, a potent inhibitor of the HIV-1 protease [9]; cytonic acids A and B, human cytomegalovirus protease inhibitors [10]; pestacin and isopestacin, antioxidants [11]; paclitaxel, antineoplastic agents [12]; and lariatins A and B, anti-HIV agents [13]. The discovery and commercial production of paclitaxel from different endophytic species revolutionized the pharmaceutical industry (with FDA approval in 1992), and the diterpene natural product garnered commercial sales of over $3 billion in 2004 [14]. 

With increasing research on the discovery of natural products from biological species, endophytes are increasingly being explored as production platforms for bioactive metabolites with diverse chemical structures. The discovery of taxol from the endophytic fungus *Taxomyces andreanae* in 1993 [15], and subsequently from other endophytic species, led to renewed research on pharmacological metabolites from endophytic sources. Furthermore, studies have established that endophytes mimic host metabolism and can produce, induce, and modify the metabolic chemical entities within hosts [16,17]. Endophytic fungi are one of the most promising candidates for endophyte-mediated natural product discovery. Diverse bioactive metabolites produced by endophytic fungi have demonstrated socio-economic importance and found applications in agriculture (as biofertilizers/biostimulants) and the environment (bioremediation), as biofuels/biocatalysts, in addition to their pharmacological attributes. Table 1 provides an overview of endophytic fungi in the environment, including multifaceted applications, key examples, and translational outcomes. As useful metabolites with multifaceted environmental implications are increasingly being discovered, different strains of endophytic fungi are being investigated and the associated limitations are being addressed for maximum use/multifaceted applications.

Although plant-associated endophytes serve as “biosynthetic platforms” of pharmaceutically important secondary metabolites, challenges exist in terms of the limited knowledge of endophyte biology and decreased production of secondary metabolites due to repeated sub-culturing of endophytic strains, projecting a need to adopt a more integrated and systematic approach towards the exploitation of endophytes in drug discovery and research. In this direction, a comprehensive insight into plant–endophyte associations and their dynamics is necessary to understand how and why endophytes biosynthesize secondary metabolites [51]. Another prospective strategy is to develop strain improvement methods using genetic engineering, optimization of culture conditions/media for cultivation, and co-culturing of different endophytic strains. Recent advances in high-throughput technologies and the “genomic revolution” have contributed considerably to natural product research and the discovery of biosynthetic gene clusters (BGCs) [52,53]. Genetic engineering of endophytes is still in its infancy and is often regarded as a progenitor of system biology and functional genomics strategies [54] with the potential for long-term translational success. Endophytes mimic their host plant in independent biosynthesis of secondary metabolites, thus acting as a prospective platform for genetic manipulation [14]. Studies have suggested the inclusion of metabolic pathways and genes in endophytes via genetic recombination between plant hosts and endophytes [55,56]. In addition, horizontal gene transfer, an evolutionary mechanism, has been suggested as an adaptive mechanism for endophytes, and it confers novel traits to the associated microbes [56]. 

Moreover, approaches in metagenomics, whole-genome sequencing, and omics biology have further enabled access to the less explored natural product reservoirs of plant-associated endophytes. The existing and emerging trends in endophytic fungi-mediated biosynthesis of novel bioactive metabolites, success stories of key pharmacological metabolites, strategies to overcome the existing/forthcoming challenges in endophytic biology, and future directions/outcomes in endophytic fungi-based drug discovery form the underlying theme of this article.

## 2. Bioactive Metabolites from Endophytic Fungi as Novel Drug Candidates

Plants possessing pharmacological properties and bioactive metabolites are used to treat ailments in modern healthcare [57]. The small molecule drugs approved from 1981–2014 comprise more than 51% of natural products [58]. However, the indiscriminate use of natural products has threatened natural resources, specifically plant species, and led to a shortage of novel bioactive metabolites. Thus, it is imperative to explore other eco-friendly alternatives to produce high-value phytochemicals and plant-associated endophytes, which are emerging as a highly attractive source. The emerging significance of secondary metabolites from endophytes is largely attributed to their multifaceted applications, ranging from pharmaceutical drugs and immune suppressants to agriculture and industrial uses. The majority (38%) of the bioactive metabolites, including antibiotics, were isolated from fungi out of 22,500 microbe-derived compounds [59]. Recently, the slow pace of antibiotic discovery and rising AMR have hampered the drug discovery process. 

Plant-associated endophytes comprise bacterial and fungal species that colonize internal plant tissues and complete their partial or entire life cycle inside the host plant. Showing universal occurrence and inhabiting almost all vascular plant species, endophytes demonstrate multifaceted advantages to the host plant by promoting growth, conferring tolerance to biotic/abiotic stresses [29], and biosynthesizing bioactive secondary metabolites [43]. Recent investigations into the ecological perspective of secondary metabolite production include protection against pathogens [60,61], mitigation of abiotic stress in plants [62], and herbivore deterrence [63]. An investigation into the natural product chemistry of endophytes highlighted the remarkable potential of endophytes in the production of phytochemicals and emerging platforms for the fabrication of novel antimicrobial entities [64]. Table 2 presents current state of knowledge on the production of endophytic fungi-mediated secondary metabolites and their pharmacological significance. It is imperative to understand plant–endophyte dynamics and how endophytes are a key source of phytochemicals. Endophytes influence their host and induce favorable responses in different ways, including elicitation of novel phytochemicals, plant growth augmentation, and conferring stress (abiotic/biotic) tolerance to the host plant [64]. 

Endophytic fungi produce diverse kinds of bioactive metabolites and are categorized as terpenoids, indole alkaloids, polyketides, and non-ribosomal peptides [136] biosynthesized via different routes. Terpenoids (composed of multiple isoprene units) are biosynthesized via the mevalonate pathway by terpene cyclases, biosynthesis of cyclic terpenes by diterpene cyclase and sesquiterpene cyclases, indole diterpenes by prenyl transferases, and carotenoid formation by phytoene synthases [137]. While indole alkaloids are biosynthesized via the shikimic pathway, polyketides by polyketide synthases from acetyl-CoA and malonyl-CoA units, and nonribosomal are formed independently of ribosomes function by NRPS enzymes, different categories of fungal metabolites originate via different biosynthetic routes [137]. The different categories of metabolites are further classified as flavonoids, coumarins, xanthones, quinones, lignans, and others and demonstrate multifaceted applications. The plethora of bioactive compounds and their analogs have displayed potential pharmacological activities in animal models and as herbal medicines against several human ailments. With scientific breakthroughs in high-throughput technologies and whole-genome sequencing, natural products from endophytes have been explored as alternative sources of novel drug-like candidates, among the 1562 FDA approved drugs, there were 64 original natural products, and 320 drugs were derivatives of natural products [5]. In addition, the development of new technologies has substantially contributed to the re-establishment of drying pipelines for novel candidates in drug discovery programs. The availability of complete genomes of endophytes and the enormous amount of information on different endophyte strains have provided remarkable insights into the molecular mechanisms of endophyte colonization and interaction with the host plant.

The bio-prospection of endophytic fungi suggests that very few fungal strains have been studied in-depth, with others highlighting the potential for the existence of enormous novel candidates. The dominant taxonomic categories for the synthesis of chemical entities are from the orders Ascomycota (~97%), Basidiomycota (~2%), and Mucoromycota (~1%), and the key metabolite-rich strains include *Fusarium*, *Aspergillus*, *Penicillium*, and *Alternaria* [59,138]. The presence of high-value (anticancer) compounds in endophytic fungi includes novel chemical entities such as terpenoids and steroids (integracides), polyketides (phomones), nitrogen-containing heterocycles (penicisulfuranols), and ester quinones [138]. Other studies have reviewed and reported the presence of metabolites for tropical diseases (including altenusin, viridiol, and cochlioquinone) [139], metabolites mimicking plant secondary metabolites (antioxidant resveratrol), antidiabetics (rohitukine), anticancer (taxol), antihypercholesteromics (lovastatin), and others [140,141]. Several other metabolites of socio-economic importance were identified from endophytic fungi between 2010 and 2017 [141,142]. Among the diverse metabolites, a few significant ones produced from endophytic fungi include saponins as nutraceuticals [143], loline alkaloids as bioinsecticides [144], chitosan as a food additive [145], taxol as a pharmaceutical [146], fatty acids as ingredients in cosmetics [145], and ascotoxin as bioherbicides [147], among others. In a remarkable study, Harpar et al. [148] predicted the relative stereochemistry of ambuic acid, isolated from the endophytic fungi, *Pestalotiopsis microspora*. The novel solid-state NMR approach led to structural elucidation and a dimeric structure was suggested for ambuic acid, consisting of a hydrogen-bonded adjacent carboxyl group, including the lattice structure details, an excellent study performed for the first time on a natural product. 

Furthermore, the quorum-sensing activity of ambuic acid in Gram-positive bacteria suggested its development as a potent antipathogenic drug for targeting the virulence expression of Gram-positive bacteria. The compound, ambuic acid, inhibited quorum sensing in bacteria via the inhibition of gelatinase biosynthesis-activating pheromone (GBAP) biosynthesis [149]. Another interesting area in bioactive metabolites from *Muscodor* spp. has gained momentum in multi-faceted applications. Diverse types of volatile organic compounds (VOCs) are produced by *Muscodor* spp. and comprise aldehydes, aromatics, esters, alcohols, terpenoids, and nitrosamides, among others [150]. The endophytic genus highlights multi-faceted attributes in industrial and agricultural applications, followed by food and healthcare, projecting direct and indirect applications. The diverse application of VOCs, isolated from *Muscodor* spp., includes agricultural applications (antimicrobial agents, for soil fumigation, biofumigation, etc.), food preservation, the perfume industry, and bioactive compounds in healthcare [150]. In addition, non-ribosomal peptides (NRP) and Penicillin (antibiotic) were isolated from a mangrove endophytic fungus, *Penicillium chrysogenum* MTCC 5108 [151]. Similarly, a polyketide synthase I (PKS I)-based screening method led to the discovery of a new polyketide, penicitriketo from endophytic fungi [152]. 

Plant-associated endophytes have gained significant momentum in bio-prospection and isolation of bioactive metabolites with multi-faceted attributes. In this direction, literature reviews have provided key insights into the existing and emerging significance of endophytes in drug discovery and research—a few articles worth mentioning include Discussion on biodiversity of endophytes and its exploitation in drug discovery [1], Bioactive metabolites, and their pharmaceutical design in drug development [2], Emerging roles and applications of actinomycetes endophytes [13], Chemical ecology of endophytes and its role in bioactive metabolite production [43], The process of horizontal gene transfer and its implications in the transfer of novel traits in production of bioactive metabolites [56], and comprehensive insight on the diverse metabolites produced by fungal endophytes and their biological functions, among others. The present review on fungal endophytes provides a detailed insight into the existing and emerging prospects of fungal endophytes as “novel candidates” in drug discovery and research and how the different yield enhancement strategies can be adopted to address the associated bottlenecks and enhance bioactive metabolite production.

## 3. Molecular Mechanisms of Plant–Endophytic Fungi Interactions

Evolutionary studies have suggested an important role of fungal partners in plant adaptation/colonization in terrestrial systems [153]. Several studies have documented the beneficial role of plant-associated endophytic fungi [29,154]; however, biological and chemical barriers need to be addressed to establish plant–fungal associations [43]. The theory of “balanced antagonism” ensures a balanced equilibrium between the adverse effects of endophyte association with plant hosts and the defense response exerted by the plants [155]. To ensure survival within plant hosts, endophytes secrete secondary metabolites that neutralize the toxic effects of plant host defense. Moreover, the invasion of endophytes into plants activates the plant defense system, limiting endophyte development and disease symptoms, if any. Asymptomatic colonization by endophytes is mediated by metabolite production, which counters host defense via multipartite symbiosis [155,156]. The phenomenon of “balanced antagonism” with the competent existing microbes is further adopted by endophytic fungi, balancing plant defense and virulence of endophytic fungi. However, if the plant host defense is overcome by endophytic fungi, it will lead to plant disease via plant–pathogen interactions [157]. A key example of this phenomenon is the production of taxol by *Paraconiothyrium SSM001* to tackle host pathogens [61]. Furthermore, the coevolution of endophytes and plant hosts has resulted in the development of sophisticated mechanisms by endophytes to modify plant immune responses [158], such as the suppression of β-glucan-triggered immunity by endophytes in multiple plants [159]. In terms of plant hosts, the evolution of defense mechanisms, including secondary metabolite production to counter pathogens, constitutes an integral host mechanism. In response to this, pathogens have developed resistance mechanisms and undergone structural modifications in some cases to counter plant defense responses [157]. An interesting mechanism displayed by endophytes within plants is promoting plant immunity by monitoring the vascular system; for example, in Pacific yew, known for taxol production by endophytic fungi, the pathogens enter the vascular system through air pockets and cracks in the bark, and the endophytes release taxol as a defense barrier to pathogens [61].

Recent advances in scientific interventions and whole-genome sequencing have contributed substantially to the bio-prospection of endophytic fungi to produce pharmacological metabolites. Endophytic fungi and their metabolic pathways are key targets for research. Although the metabolic pathway has defined a genetic framework to produce a specific metabolite, gene clusters mostly remain silent under laboratory conditions [3]. Moreover, strategies to activate silent gene clusters through co-cultivation of different endophytic fungal strains, mutagenesis, and genetic engineering have been adopted to improve the biosynthetic potential of different endophytic strains, with limited success. Moreover, the bioactive metabolites isolated from endophytic fungi are classified into different categories, including phenolics, flavonoids, saponins, alkaloids, terpenes, and xanthones [160], demonstrating multiple pharmacological properties. However, the decreased production due to repeated sub-culturing of the microbial strains leads to an unstable yield in axenic cultures, which accounts for the few existing challenges in endophyte biology and research. Different researchers have proposed parallel hypotheses on the genetic origin of metabolites (both in endophytes and plants), including the presence of parallel biosynthetic pathways for metabolite production in plants and endophytes. However, examples of taxol gene clusters showed low similarities between plants and microbes [161], suggesting an independent evolution of biosynthetic pathways in different organisms [162]. Another key example is camptothecin (CPT) production by *Fusarium solani* utilizing the host enzyme strictosidine synthase for CPT biosynthesis, suggesting the horizontal transfer of gene clusters between endophytic fungi and host plants in an evolutionary course [163]. Both endophytic fungi and plants possess resistance mechanisms against CPT and taxol, respectively. Another independent hypothesis claims that host mimicking by endophytic fungi leads to the production of bioactive compounds, such as plant hosts, questioning the actual biosynthesis by endophytic fungi [43]. However, limited knowledge about the molecular mechanisms of plant–associated endophytic fungi is a major challenge in the large-scale commercial production of bioactive metabolites from endophytic fungi.

## 4. Scientific Approaches for Natural Product-Based Drug Discovery

Key strides have been made in the discovery and characterization of valuable bioactive metabolites from natural sources. Substantial efforts to identify novel bioactive metabolites have opened new avenues in natural product-based drug discovery from endophytic fungi. The increasing importance of natural products in pharmacological applications has greatly impacted the large-scale production of high-value metabolites. Plant–microbe associations, particularly endophytic fungi, continue to intrigue researchers worldwide with a considerable potential to impact the pharmaceutical industry [164,165]. With the alarming rise in AMR and declining drug pipelines, novel metabolites from natural sources are increasingly being studied for their therapeutic potential. In this direction, advances in high-throughput technologies and scientific interventions have provided firm grounds as “discovery platforms” in natural product-based drug discovery programs. In addition, the emerging revolution in interdisciplinary strategies comprising computational methods and multi-omics strategies has redefined natural product discovery from plant–endophytic associations [166]. Figure 2 provides a schematic overview of the traditional and emerging scientific approaches for endophyte-based drug discovery.

## 5. Traditional Scientific Methods

Recently, there has been an increased exploration of plant–endophytic associations for the discovery of high-value metabolites with potential pharmacological applications. Deep learning methods have defined an interesting platform for the bio-prospection of endophytes, regulatory networks, and the prediction of novel chemical entities [166]. As discussed earlier, plant-associated endophytes have defined an attractive “biosynthetic platform” for the synthesis of novel bioactive metabolites; however, several challenges exist in harnessing these biological sources in drug discovery. For example, most of the secondary metabolite pathways are silenced, and the knowledge of metabolic networks and mechanisms needs to be understood in detail. Furthermore, in silico approaches predict the existence of a wide array of metabolites that remain silent/inactive under native conditions (in planta) [167,168]. In recent decades, natural product discovery from endophytes relied on powerful, low-throughput methods [3], essentially including culture-based methods, plant-based extraction methods, biochemical screening/isolation using HPLC, NMR, and MS, and bioactivity-guided isolation [169], with existing limitations. Toward addressing the existing challenges, sophisticated methods, namely the “one strain-many compounds” (OSMAC) approach, engineering ribosomes, heterologous expression of genes, and promoter studies have been widely explored [170]. The OSMAC method is based on studying a particular strain and its growth under different culture conditions to produce a diverse array of metabolites [168]. These promising tools enhance gene expression in endophytes via biotic/abiotic triggering mechanisms. 

Another interesting strategy is the co-cultivation of different endophytic fungal strains to elicit gene expression in silent gene clusters [171]. For example, gene expression for taxol production was reclaimed in *Aspergillus terreus* when co-cultivated with *Podocarpus gracilior* leaves [172]. Furthermore, genetic methods have been successful in activating silent BGCs and include inducible/constitutive promoters [173], host ribosome engineering [174], and mutant selection [175]. In addition, for culturable endophytic bacteria, some important high-throughput techniques used include high-throughput elicitor screening with imaging mass spectrometry, with the potential to induce silent gene clusters [176], and most techniques were effective for culturable microbes only.

## 6. Deep Learning Approaches

One of the key approaches for analyzing large data comprises artificial intelligence, further classified as deep learning and machine learning. These methods predict the distribution pattern of the plant microbiome (environmental niches) and may accurately predict the biosynthesis of bioactive metabolites from endophytes [177], highlighting a potential approach to replace challenging global, comprehensive estimation. In this approach, the initial dataset could be obtained from the increasing multi-omics and genomics combined with plant metabolomics data. The targeted region can then be analyzed through multi-omics methods, further co-integrated with metabolic pathway analysis [178,179,180]. Currently, multiple deep learning and machine learning strategies are employed to improve drug discovery from natural products. The software DeepBGC relies on a word2vec-like word embedding skip-gram neural network and bidirectional long short-term memory (BiLSTM) neural network, subject to a large dataset from microbial communities [181]. Furthermore, bio-prospection of endophytes for bioactive metabolites may be performed by integrating genome metabolic models and deep learning methods. Similarly, for the diversity prediction of chemical entities, deep learning chemi-informatics methods may be employed for efficient prediction. In addition, computational biology methods emphasize the analysis of chemical entities and metabolomics strategies, targeting a predictive model of plant microbiome and associated biochemical changes. In keeping with the target goal, the abovementioned deep learning methods may be suitably altered; for example, the objective may be to find chemical novelty or a target function (antimicrobial) or elucidation of complex structures [166]. The established genome mining tools aim to understand metabolic flux and pathway regulation, regulatory processes, and metabolite interactions. With the enormous amount of data generated from high-throughput experiments, the OSMAC tool, co-integrated with metabolomics information, forms a basis for computational analysis. These novel computational pipelines project revolutionary landmarks in natural product-mediated drug discovery from endophytes. 

## 7. High-Throughput Strategies

Advances in computational biology and whole-genome sequencing have been instrumental in defining new avenues for endophyte-mediated natural product discovery. Comparative genomics aims to understand the diversity of chemical entities from microbes because of the conserved BGCs across species, associated with regulatory genes, uptake, and product transport [166]. Moreover, BGCs, which are specific, are mostly silent/lowly expressed under laboratory conditions, making functional gene prediction difficult. An effective approach toward the discovery of natural products begins with in silico predictions through genome information and proceeds to experimental validation via activation of the biosynthetic pathways [178]. The ongoing efforts in the development and upgradation of computational resources have greatly refined research on natural product-mediated drug discovery. Databases such as the Database of BIoSynthesis clusters CUrated and InTegrated (DoBISCUIT) [182], and ClustScan software [183], and ClusterMine 360 [184] have facilitated the discovery/identification of novel gene clusters. Moreover, the discovery of BGCs via phylogenetic analysis was performed through a useful database, The Minimum Information on Biosynthetic Gene clusters (MIBiG) [185]. Other bioinformatics databases promoting the annotation of BGCs include Reconstruction, Analysis and Visualization of Metabolic Networks’ RAVEN 2.0 software [186], antibiotics and secondary metabolite analysis shell (antiSMASH) [187], and Metaflux [188], and the genome-wide identification and profiling of gene clusters of natural products have become attainable. In addition, bioinformatics software (based on the network algorithm) improves predictions of genome mining methods [189] and may be co-integrated with metabolic modeling approaches [190]. The predictive strategies further extend the understanding of metabolic interactions in microbial cultures [191], bio-kinetic models for the estimation of interspecific interactions among microbes [192], single-cell analysis for endophyte metabolism [192] and transcriptome approaches for comprehensive understanding [193].

## 8. Pharmacological Metabolites: Case Studies, Mechanism of Action, and Commercial Success

Medicinal plants form the backbone of the traditional system of medicine and are a rich source of pharmacological metabolites for the treatment of diseases and prospects as modern medicines [56]. Since the initial reports of taxol production from the endophyte *Taxomyces andreanae*, similar to its plant host (*Taxus brevifolia*) in 1993, endophytes have been increasingly explored for the production of high-value metabolites. Plant-associated endophytes, particularly endophytic fungi, have emerged as production platforms for pharmacological metabolites with diverse therapeutic applications [2]. There has been an upsurge in the production and marketing of pharmacological metabolites worldwide, and a few remarkable examples and their success stories in the pharmaceutical sector have been discussed. Table 3 presents key examples of commercially available drugs from endophytic fungi: key examples, pharmacological functions, bottlenecks, and success stories.

## 9. Taxol Production

The anticancer drug taxol is one of the most successful drugs marketed globally for its anticancer activity. Taxol was isolated from the medicinal plant *Taxus brevifolia*, but its concentration was too low (0.001–0.05%) in most species. Therefore, 1 g of taxol production requires 15 kg of tree bark, while the anticancer dose amounts to 2.5 g [208]. To increase taxol production, alternative sources have been investigated to meet the rising therapeutic demands. The bioactive metabolite, taxol was first isolated from the endophytic fungi *Taxomyces andreanae* [209], and subsequently from other endophyte species. The discovery (in 1960), structural elucidation (1971), and FDA approval (1992) of taxol defined new paradigms in global commercial success, projecting a global market of USD 78.77 million in 2017. In chemotherapies for cancer treatment, paclitaxel is a mitosis inhibitor and is widely used for the treatment of cancers such as cervical, ovarian, and breast cancers. The global consumption trends indicated a market share of 53% (2017), with Europe accounting for 19% of global market trends [210].

Taxol and its derivatives represent a popular class of anticancer drugs produced by different endophytes. The mechanism of action includes inhibition of mitosis and promotion of tubulin depolymerization during cell division [211]. After several research attempts, the taxol-producing endophyte *T. andreanae* was discovered in *Taxus brevifolia*, detected via electrospray mass spectroscopy [212] and compared with the standard paclitaxel. The identification of taxol and its anticancer potential was bright, but the existing crisis and the low content in the plant projected major concerns to increase supply. Subsequently, paclitaxel-producing endophytes have been reported (taxol-producing endophytes from *Taxus baccata* L.), characterized, and validated using HPLC-MS [213]. The five taxol-producing endophytic fungi were *Fusarium redolens*, *Gibberella avenacea*, *Fusarium tricinctum*, *Paraconiothyrium brasiliense*, and *Microdiplodia* sp. G16A, with the highest taxol yield of 66.25 μg/L by *Fusarium redolens* [214]. The diterpenoid metabolite was further isolated from several endophytes belonging to the Ascomycota, Basidiomycota, Deuteromycota, and Zygomycota classes [215,216]. However, successful translational research towards industrial production faces certain challenges currently, including decreased production because of repeated sub-culturing and low concentration in biological species. Comprehensive knowledge of endophyte biology and dynamics is essential for improving the commercial production of important bioactive metabolites [2,217].

Presently, considering the global market for anticancer drugs, research efforts have been made for large-scale production of taxol from endophytic fungi [218,219,220]. Fermentation methods have been optimized for taxol production by supplementing nitrogen/carbon sources, precursors, and elicitors [221] and defined conditions related to temperature, pH of culture, and dissolved oxygen [222]. Moreover, inhibition of the metabolic pathway of sterols (ergosterol) by triadimefon (inhibitor) is effective in enhancing the production of paclitaxel [223]. However, taxol production in endophytic fungal cultures leads to the reprogramming of fungal physiology and the regulatory loss of metabolite production [141]. Co-cultivation of *Paraconiothyrium* sp. with *Alternaria* sp., and *Phomopsis* sp. enhanced taxol production eightfold [224]. Genetic transformation of endophytic fungi is an emerging method for increasing the production of targeted metabolites [220]. Protoplast-mediated transformation [225], *Agrobacterium*-mediated transformation [226], electroporation [227], and genome editing via CRISPR-Cas [228] are some other scientific approaches employed to enhance taxol production by endophytic fungi.

## 10. CPT Production

CPT is a monoterpene indole alkaloid that is commercially marketed as an anticancer drug. Initially, it was isolated from the stem and bark of *Camptotheca acuminata*, a native Chinese tree. CPT has a pentacyclic structure and is a topoisomerase inhibitor that binds to topoisomerase I and DNA complex and stabilizes it, leading to DNA damage and apoptosis (https://en.wikipedia.org/wiki/Camptothecin, accessed on 23 July 2021). In clinical trials, CPT showed anticancer activity against colon, lung, breast, and stomach cancers. The commercial success of CPT is attributed to its applications in cancer chemotherapies, and its derivatives are marketed with the name belotecan, topotecan, irinotecan, and trastuzumab deruxtecan [229,230]. CPT was commercially isolated from *Nothapodytes nimmoniana* and *C. acuminata* and has a high global demand for cancer treatment [231]. Although some CPT derivatives are commercially marketed (irinotecan, belotecan, and topotecan), others such as diflomotecan, tenifatecan, genz-644282, chimmitecan, exatecan, lipotecan, silatecan, cositecan, and simmitecan are under clinical trials [232,233]. The increasing demand for CPT has led to indiscriminate use/overharvesting of the two plants, making them an endangered species [234]. To produce 1 ton of CPT, approximately 1000–1500 tons of plant wood chips are required [235], leading to excessive exploitation of the plant species. 

The emerging popularity of endophytes as production platforms for high-value metabolites necessitates the potential bio-prospection of endophytes for natural product drug discovery. CPT-producing endophytes have been discovered, including *Trichoderma atroviride* LY357 [236], *Fusarium solani* strain ATLOY-8 [237], and *Neurospora* sp. [238]. However, the variable/inconsistent production, yield loss because of sub-culturing, and low concentrations are the major bottlenecks in the commercial-scale production. Mohinudeen et al. [239] reported the presence of a high-CPT-producing endophyte *Alternaria* sp. from *N. nimmoniana*. The endophyte produced up to ~200 μg/g of CPT in axenic cultures and demonstrated cytotoxic activity against cancer cell lines [239]. Regarding the molecular pathways in CPT production, strictosidine synthase, tryptophan decarboxylase, secologanin synthase, and geraniol 10-hydroxylase are some of the key enzymes identified to date. Kusari et al. [43] identified tryptophan decarboxylase, secologanin synthase, and geraniol 10-hydroxylase in endophytic fungi. The absence of the strictosidine synthase gene in endophytes has led to the presumption of the involvement of host genes in CPT biosynthesis. Decreased CPT yield due to sub-culturing was explained by the absence of selection pressure, leading to the degradation of biosynthetic pathways in the endophytic fungi [41], limiting CPT biosynthesis. Recent advances employing endophytes as production platforms for CPT have focused on isolation/fermentation of CPT-producing endophytes, extraction, and detection via different methods, and employing fermentation culture to improve CPT yield [240].

## 11. Vinca Alkaloids (Vincristine/Vinblastine) Production

Vinca alkaloids are pharmacological metabolites isolated from Madagascar periwinkle (*Catharanthus roseus*) and are listed on the World Health Organization’s List of Essential Medicines [241]. Vinca alkaloids are used as effective medications for the treatment of different types of cancers, including bladder cancer, melanoma, and lung cancers (https://en.wikipedia.org/wiki/ accessed on 23 July 2021). Vinblastine is an analog of vincristine, and its anticancer mechanism is defined by binding to tubulin, thereby inhibiting microtubule assembly and formation, and is regarded as an effective chemotherapeutic agent. Two of the most successful commercial drugs—vinblastine and vincristine—are present at very low concentrations in plants. Moreover, the estimated global demand for bioactive metabolites (0.3 tons annually) and global market of USD 200 million projects additional requirements from alternative natural resources [242]. According to Balandrin and Klocke [243], 500 g of *C. roseus* leaves is required to produce 1 g of pure vincristine. Although vinca alkaloids differ slightly in their chemical structure and mechanisms, their clinical activity and toxicity vary. The commercial success and importance of vinca alkaloids have led to studies on endophytes from *C. roseus*. Several endophytic fungi were isolated and screened to produce vincristine/vinblastine, and a few key examples are *Aspergillus*, *Alternaria*, and *Cladosporium* sp. Moreover, the endophytic fungus *Talaromyces radicus* (from *C. roseus*) produced vincristine in substantial amounts (670 μg/L) in modified M2 medium and vinblastine in PDB medium (70 μg/L). Furthermore, vincristine was purified, and it demonstrated cytotoxic activity against cancer cell lines [244].

## 12. Podophyllotoxin Production

Another prospective pharmacologically important metabolite podophyllotoxin, an aryl tetralin lignin, is widely present in angiosperms and gymnosperms [245]. Podophyllotoxins are emerging as pharmacological metabolites because of their antiviral and cytotoxic activities [246,247]. Some semisynthetic derivatives of podophyllotoxin, teniposide, and etoposide have been approved for the treatment of leukemia and different types of cancers [248]. Because of its low concentration in plants, etoposide was developed and approved by the FDA in 1983. Other analogs were chemically synthesized but had a low yield. Research efforts to enhance the production of podophyllotoxin and its analogs from different species have been conducted. The plant tissue culture method is a reliable and sustainable approach to produce bioactive metabolites from natural sources [249]. The mechanism of action of teniposide and etoposide is marked by interaction with topoisomerase II [250] by two mechanisms: it exerts its action by enhancing topoisomerase II DNA covalent complex levels or the removal of topoisomerase II catalytic function [251]. Moreover, DNA breaks induced by enzymes promote sister chromatid exchange, recombination, translocation, and insertions/deletions [252]. 

Different strains of endophytic fungi have been studied for their potential to biosynthesize podophyllotoxin, and *Phialocephala fortinii*, endophytic fungi from *Podophyllum peltatum*, produced podophyllotoxin in the range of 189 and 0.5 μg/L. Furthermore, the cytotoxic potential of the endophytic fungi was evaluated, and the LD_50_ values were 2–3 μg/mL [200]. Another species, *Fusarium oxysporum*, isolated from *Juniperus recurva*, produced 28 μg/g of dry mass of the metabolite [253]. The bioactive metabolite has been detected/produced by several species of endophytic fungi, a few prominent ones comprise *Aspergillus fumigatus* from *Juniperus communis* [248], *Trametes hirsuta* from *Podophyllum hexandrum* [45], and *Mucor fragilis* [254]. The ongoing bio-prospection of different endophytic species for pharmacological metabolites highlights its existing and emerging importance in natural product-mediated drug discovery.

## 13. Existing/Potential Bottlenecks in Biotechnological Applications

Endophytic fungi have great potential to impact the pharmaceutical industry, subject to the bio-prospection of novel pharmacological metabolites with multifaceted implications. However, little is known about plant–endophyte dynamics, low yield, and loss of yield because sub-culturing and scaling-up cultures constitute some key bottlenecks. An emerging demand and constant supply of novel, high-value metabolites define global trends, considering the drying pipeline of antibiotic arsenals. Although the plant–endophyte interaction remains poorly understood, less information on endophytic interactions with other microbes further hampers endophyte applications. Genetic instability in plant tissue culture, slow fungal growth, and maintenance issues have decreased the use of tissue culture methods [255]. In this regard, fermentation technologies are an attractive platform to produce high-value metabolites from fungal cultures. These methods offer distinct benefits in terms of less expense, rapid growth, and optimization of culture conditions for effective production of the targeted metabolite, and sustainable production is achieved. The factors in scale-up processes, namely, oxygen solubility, viscosity of the medium, temperature, time of cultivation, pH, and others, and their optimization, make the process cumbersome. The culture conditions need to be optimized and regulated properly for the large-scale targeted production of metabolite of interest [168]. Genetic engineering approaches for the activation of BCGs for enhanced production hold potential; however, with very few reported studies on endophyte chassis, the genetic manipulation methods need to be further optimized. The combinational synthesis for complex metabolites highlights associated problems and may be resolved by employing elicitors for increased production of metabolites in a particular metabolic pathway. Other challenges include the complex structure of metabolites and undesired pathway intermediates, resulting in the inhibition of enzyme function. The existing areas of concern include the adverse effects of endophytic fungi in the control of pathogens. For example, endophytes of *Crocus sativus* may be pathogenic to host plant and result in infection, as endophytes are considered latent pathogens capable of causing diseases [256]. A deep understanding of the associated risk factors with endophytes is necessary to avoid adverse effects and benefit biocontrol applications.

## 14. Strategies for Yield Enhancement of Pharmacologically Important Metabolites

Substantial efforts to isolate and screen endophytes for high-value metabolites resulted in undesirable/low yields of the target metabolite. Although novel chemical entities such as CPT, paclitaxel, podophyllotoxin, and huperzine A were isolated from endophytic fungi, bottlenecks with large-scale production and commercialization remains a major concern. As the field of endophytes is gaining popularity in multifaceted environmental applications, scientific tools and approaches are aimed at delineating the dynamics of plant–endophyte interactions and enhancing the yield of pharmacologically important metabolites. Co-integration of bioprocess techniques and genetic/metabolomics approaches may be employed to effectively target and produce high yields of the desired metabolites.

## 15. Metabolic Engineering of Endophytic Fungi

Studies on the engineering of endophytes are preliminary and investigate endophyte chassis to improve the yields of targeted metabolites through genetic strategies. Genetic manipulation of endophytes defines potential future outcomes by introducing key pathway genes via genetic transformation for yield enhancement of high-value metabolites in axenic cultures. Studies on the genetic transformation of taxol-biosynthesizing endophytes include the overexpression of genes and genome rearrangement with mutagenesis for enhanced metabolite production [257,258]. Random mutagenesis approaches for genetic modification of the endophyte genome and induction of metabolites and mutant screening have been adopted [258]. Taxol production was increased (endophyte strain HDF-68) through protoplast fusion of two strains—UL50-6 and UV40-19—leading to a 20–25% increase in taxol yield [259]. Wang et al. [260] showed that the endophytic fungus *Phomopsis* sp. produces deacetylmycoepoxydiene (DAM) (antitumor metabolite), and genome shuffling of eight parental protoplasts resulted in a high-yielding strain (produced >200-fold DAM) in the transgenic endophytic strain [260]. Similarly, previous studies on *Pestalotiopsis microspores* (endophytic fungi) aimed to decode the taxol biosynthetic pathway via protoplast transformation [261]. These unexpected results led to the detection of extrachromosomal DNA in the transformants, suggesting a key role in development and adaptation. The translational success and marketing of taxol, led to extensive investigations on taxol-producing endophytes: with the application of genetic engineering defining a key platform in endophyte chassis; some important examples include, restriction enzyme-mediated integration in *Ozonium* sp. strain BT2 (taxol-producing fungi) [262]; *Agrobacterium*-mediated genetic transformation in *Ozonium* sp. EFY21, with enhanced efficiency of transformation [263] and PEG-mediated transformation of *Ozonium* sp. EFY-21 [264]. In this direction, multiple approaches for genome modification/shuffling of endophytes have been attempted to increase the metabolic flux, either by limiting competitive pathways or by reorientation of precursor supply, strain improvement methods leading to yield enhancement of targeted metabolites. In an interesting study of pathway reconstitution, the taxadiene-biosynthetic cluster was expressed in *Escherichia coli* and transformed into *Alternaria alternata* TPF6 for taxadiene production. This study was able to address the challenges associated with the first step of taxadiene production, and a consistent supply of taxadiene in *A. alternata* TPF6 (61.9 ± 6.3 μg/L) was obtained [265]. The inactivation of the sterol biosynthesis pathway improved the taxol yield dramatically, suggesting the prospects of genetic engineering as a means of yield enhancement of metabolites. Genome editing of filamentous fungi using the CRISPR-Cas system promises to revolutionize the production of natural products. The CRISPR-Cas9 tool aims at Cas9 gene delivery and guides DNA in cassettes and formation of active Cas9/guide RNA complex and requires markers, shuttle vectors, and promoter sequences for efficient expression. The validation of BGCs and targeted metabolite production may be improved by employing the CRISPR-Cas genome editing approach for the transformation of endophytic fungi.

## 16. Mutagenesis of Endophytic Fungi

To exploit endophytes on a commercial scale, it is imperative to understand and tap into the biosynthetic potential of endophytes to enhance the yield of high-value metabolites. To address the concern toward low-yielding endophyte strains, strain improvement strategies may be employed to enhance yield and other characteristics, including utilization of nitrogen/carbon sources, changes in morphology, and reduction in undesired metabolites [266]. In addition to genetic manipulations, mutagenesis of endophytes by protoplast fusion or random screening offers distinct advantages and prospects in this direction. The genetic framework of microbial strains is altered by employing mutagens and comprises either chemical (nitrosoguanidine, diethyl sulfate, ethyl methyl sulfonate) or physical (X-ray, γ-rays, microwave, etc.) mutagens. Moreover, mutant strains were analyzed using random methods to identify mutants with a particular genotype of interest [267]. Mutations are also induced by using two or more mutagens or one repeatedly [220]. Mutagenesis of endophytic fungi leads to regulatory gene alterations, causing phenotypic changes and enhanced secondary metabolite production, although the process needs to be defined [268]. A key example by Kai et al. [269] discussed the mutagenesis of endophytes and the selection of hygromycin-resistant strains for enhanced taxol production. In endophytic fungi, mutagenesis is attempted by employing mycelia, spores, or protoplasts and applied to taxol-producing endophytes with remarkable success [242]. Although mutagenesis offers a prospective approach, mutation of endophyte mycelium highlights the demerits of genetic divergence in the offspring, and the optimization of conditions for endophyte spore mutagenesis is difficult, suggesting that protoplast fusion is the best way to induce mutagenesis [269]. Studies on mutagenesis of endophytes for strain improvement/yield enhancement focused on optimal conditions for protoplast preparation, optimizing factors such as time, pH, temperature, enzymes, medium, and culture conditions [270,271,272]. A few reports have discussed the yield enhancement of the targeted metabolite via protoplast mutagenesis [269,273] highlighting its beneficial outcomes. Zhao et al. [271] reported the generation of an improved taxol-producing strain (HDF-68) through the fusion of two mutant protoplasts of *Nodulisporium sylviforme*. The taxol production in the mutant strain increased to 468.62 μg/L compared to the parent endophyte strains, suggesting mutagenesis as a powerful approach for yield enhancement, with some considerations. 

## 17. Co-Culture of Different Endophyte Strains

With the discovery of many endophytes and bio-prospection of their respective strains, the compatibility of endophytes to produce a target metabolite has been explored [153]. The individual endophyte strains are maintained or preserved as monocultures and cultured to produce high-value metabolites; however, the standard culture may not hold the potential to activate the expression of BCGs in endophytic fungi, leading to a decrease in the discovery of novel metabolites. Sequencing of whole genomes showed that multiple genes encoding bacterial/fungal enzymes surpassed the secondary metabolites in these microbes [274]. Moreover, the metabolic processes in microbes are significantly changed with little fluctuation in culture conditions, specifically in endophytes with host-balanced interactions and multipartite interactions [242]. The co-cultivation of different endophytic strains aims to activate some silent BGCs in the presence of compatible microbes and enhances the production of targeted metabolites of interest. This is relevant in the case of antibiotic production when competition governs the biosynthesis of a product in cultures [157]. *Corynespora cassiicola* SUK2 and *Colletotrichum fructicola* SUK1 (endophytic fungi), isolated from *Nothapodytes nimmoniana* (Grah.) Mabb. (Ghanera), demonstrated host-independent biosynthesis of CPT under fermentation conditions, with a significantly higher yield (>1.4-fold) than that of monocultures [275]. Ahamed and Ahring [257] co-cultivated *E. coli* with *Gliocadium roseum* (endophytic fungi) and increased hydrocarbon production by 100-fold, compared to pure cultures of *G. roseum*. Other key examples, namely co-cultivation of *Phomopsis* and *Alternaria*, led to an eightfold increase in taxol production [224], and *Alternaria* and *Paraconiothyrium* SSM001 co-cultures led to a threefold increase in taxol production, and other examples suggest that co-culturing of compatible endophyte strains highlights an attractive alternative for yield enhancement of high-value metabolites. 

## 18. Optimization of Culture Conditions

Although the production of diverse chemical entities by endophytic fungi is prospective in addressing natural product-mediated drug discovery, the multiple factors that govern production remain poorly understood. The cultural factors that play a key role in metabolite production include temperature, pH, cultivation time, nutrients, and aeration in the medium [276]. The term OSMAC was coined by Bode et al. [277] to show that a single microbial strain may biosynthesize multiple compounds by changing culture conditions, leading to the discovery of high-value metabolites [104]. Optimization can be achieved by considering a one-factor change or multiple-factor variations for fermentation and medium optimization. In addition, the statistical design provides reliable information for experiments by enabling the analysis of experimental variables and making an accurate prediction [278]. Wang et al. [279] discussed the importance of optimization of factors (such as carbon source, pH, and temperature) for *Neurospora intermedia* DP8-1 (diuron-degrading endophyte) from sugarcane and reported effective biodegradation of up to 99% of the diuron present. In the endophytic fungi *F. solani* from *Ferocactus latispinus*, the pH value and nitrogen/carbon ratio were found to affect polyketide production and were optimized, resulting in a higher yield (476 μmol/L) [280]. In addition, a bioreactor was developed to monitor the consistent production of metabolites [281]. The colonization of plant internal tissues by fungal endophytes (dark condition) showed that light treatment decreased taxol production in *Paraconiothyrium* SSM001 and decreased the level of gene expression [282].

## 19. Epigenetic Modifiers

The BCGs in fungi are present in the heterochromatin state and are controlled by epigenetic processes, including DNA methylation and histone deacetylation [283]. Gene expression/silencing in fungi is regulated by chromatin modification via DNA methylation/histone deacetylation. Epigenetic modifiers such as histone deacetylase (HDAC) inhibitors have the potential to initiate remodeling of chromatin to activate BGCs [284]. In addition, 5-aza-20-deoxycytidine and 5-azacytidine (DNA methyltransferase) inhibitors are another important class of epigenetic modifiers used for the discovery of natural products. However, an in-depth understanding of secondary metabolite biosynthesis and regulation is essential for the development of new techniques for the isolation of secondary metabolites from fungi [242]. Prospective approaches for discovery/isolation of high-value metabolites include BCG-specific transcription factor overexpression, promoter exchange, or employing inducible promoters for gene activation, and applied to endophytic fungi to alter secondary metabolite networks. 

## 20. Translational Success and Outcome of Endophyte Research

The discovery and commercial production of taxol was a landmark in endophyte biology and research and marked the bio-prospection of endophytes in natural product-mediated drug discovery. The key pharmacological metabolites, including CPT, podophyllotoxin, huperzine A, and vinca alkaloids (vincristine/vinblastine), define a novel paradigm in pharmacology, with a new hope to discover and employ high-value metabolites in therapeutic endeavors. Although endophytes continue to gain popularity as an attractive production platform for pharmaceutical therapeutics at an affordable scale, the existing and emerging bottlenecks regarding the purity of endophyte strains, sub-culturing concerns, low yields, and adverse effects need to be further addressed for maximal utilization. Recent advances in scientific interventions and high-throughput methods have made substantial contributions to endophyte research, aiming to address the projected challenges. Genetic engineering of endophytes comprising gene overexpression and gene constructs, strain mutagenesis, and co-cultivation of compatible strains is targeted toward the improvement of different endophyte strains and high yield of targeted metabolites. With the depletion of natural resources and limited availability of natural products, the bio-prospection of endophytes and their genetic improvement is an important area of research in drug discovery programs.

## Figures and Tables

**Figure 1 microorganisms-10-00360-f001:**
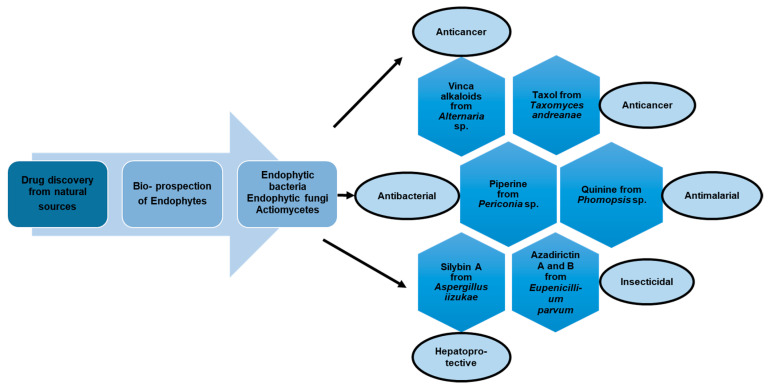
Bio-prospection of endophytes and discovery of novel, high-value metabolites of commercial significance.

**Figure 2 microorganisms-10-00360-f002:**
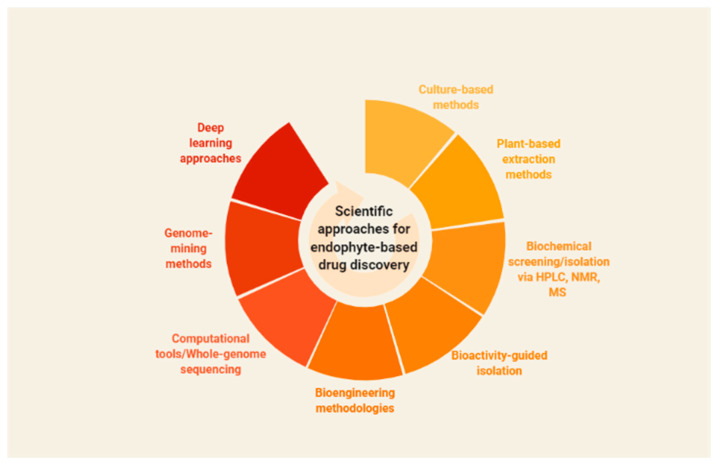
Provides a schematic overview of the traditional and emerging scientific approaches for endophyte-based drug discovery (Created with BioRender.com, accessed on 24 January 2022).

**Table 1 microorganisms-10-00360-t001:** Endophytic fungi in environment: significant multi-faceted applications, key examples, and translational outcomes.

Biological Application	Endophytic Fungi	Plant Species	Outcome	Reference
Plant growth promotion and agriculture				
Plant growth promotion (PGP)	*Penicillium* sp. 21	*Camellia sinensis*	Mineral-solubilizing function (Ca_3_(PO_4_)_2_ and rock phosphate)	[18,19]
	*Penicillium* sp. 2			
	*Aspergillus* sp. MNF			
	*Trichoderma gamsii* (NFCCI 2177)	*Lens esculenta*	Solubilization of Tricalcium phosphate	[20]
	*Trichoderma peudokoningi*	*Solanum lycopersicum*	Siderophore production, HCN and ammonia production	[21]
	*Chaetomium globosum*			
	*Fusarium oxysporum*			
	*Ophiosphaerella* sp.	*Triticum aestivum*	PGP activities	[22]
	*Cochliobolus* sp.			
	*Cladosporium sphaerospermum*	*Glycine max*	Solubilize calcium phosphate	[23]
	*Fusarium tricinctum* RSF-4L	*S. nigrum*	Production of phytohormones (gibberellins)	[24]
	*Alternaria alternata* RSF-6L			
	Endophytic fungi	*Hordeum murinum* subsp. *murinum*	IAA production, PGP activities Plant yield increase	[25]
Biofertilizers/biostimulants for crops	*Aspergillus* sp.	*S. officinarum*	Phosphorous solubilization	[26]
	*Penicillium* sp. 1			
	*Penicillium* sp. 2			
	*Cochliobolus* sp.	*T. aestivum*	Phosphorous solubilization	[26]
	*Curvularia* sp.			
	*Fusarium equiseti*	*Pisum sativum*	Phosphorous solubilization	[27]
	*Coniothyrium aleuritis isolate 42*	*Lycopersicon esculentum*	Plant biomass increase, fruit yield	[28]
	*Pichia guilliermondii* isolate F15			
	*Fusarium oxysporum* strain NSF2			
	*Fusarium proliferatum* strain AF04			
	*Aspergillus nidulans* strain FH5			
	*Trichoderma spirale* strain YIMPH30310			
Biocontrol function	*Fusarium verticillioides*	*Zea mays*	Restrict *Ustilago maydis* growth	[29]
	*Penicillium* sp.	*Cucumis sativus*	Biocontrol of *Fusarium oxysporum* f. sp. *cucumerinum*	[30]
	*Guignardia mangiferae*			
	*Hypocrea* sp.			
	*Neurospora* sp.			
	*Eupenicillium javanicum*			
	*Lasiodiplodia theobromae*			
Biotic/abiotic stress tolerance	*Piriformospora indica*	*Hordeum vulgare*	Drought stress tolerance	[31]
	*P. indica*	*Brassica rapa*	Drought stress tolerance	[32]
	*P. indica*	*H. vulgare*	Biotic/abiotic stress tolerance	[25]
Bioremediation	*Mucor* sp. MHR-7	*Brassica campestris*	Metal toxicity reduction	[33]
	*Rhizopus* sp. CUC23	*Lactuca sativa*	Chromium detoxification	[34]
	*A. fumigatus* ML43			
	*Penicillum radicum* PL17			
	Endophytic fungi	*Agrostis stolonifera*	Bioremediation of lead	[35]
	*Neotyphodium coenophialum*	*Festuca arundinacea*, *Festuca pratensis*	Bioaugmentation, total petroleum hydrocarbons (TPH) and polycyclic aromatic hydrocarbons (PAHs) removal from the soil	[36]
	*N. uncinatum*			
	*Verticillium* sp. *Xylaria* sp.	Plants from Ecuadorian Amazon	Degradation of Petroleum hydrocarbon	[37]
	Endophytic fungi	-	Bioremediation of synthetic plastic polymers	[38]
	*Curvularia* sp.	*Mangrove* sp.	Heavy metal biosorption	[39]
	*Neusartorya* sp.			
	*Bjerkandera adusta* SWUSI4	*Sinosenecio oldhamianus*	Detoxification of triphenylmethane dyes	[40]
	*Lasiodiplodia theobromae*	*Boswellia ovalifoliolata*	Heavy metal tolerance	[41]
	Lindgomycetaceae P87 *Aspergillus* sp. A31	*Aeschynomene fluminensis*	Heavy metal resistance, bioremediation	[42]
Bioactive metabolites for industrial and pharmacological applications				
Paclitaxel	*Taxomyces andreanae* *T. brevifolia*	Pacific yew	Anticancer	[15]
Azadirachtin A and B	*Eupenicillium parvum*	*Azadirachta indica*	Insecticidal	[43]
Subglutinol A	*Fusarium subglutinans*	*Tripterygium wilfordii*	Immuno-suppressant	[44]
Isopestacin	*Pestalotiopsis microspora*	*Terminalia morobensis*	Antifungal, Antioxidant	[11]
Podophyllotoxin	*Trametes hirsuta*	*Podophyllum hexandrum*	Antiviral, Radio-protective	[45]
Forskolin	*Rhizoctonia bataticola*	*Coleus forskohlii*	Anti-HIV, Antitumor	[46]
Sanguinarine	*Fusarium proliferatum*	*Macleaya cordata*	Antihelmintic	[47]
Digoxin	*Alternaria* sp.	*Digitalis lanata*	Cardiotonic	[48]
Quinine	*Phomopsis* sp.	*Cinchona ledgeriana*	Antimalarial	[49]
Capsaicin	*Alternaria alternata*	*Capsicum annuum*	Cardio-protective	[50]

**Table 2 microorganisms-10-00360-t002:** Production of endophytic fungi-mediated secondary metabolites and their pharmacological significance.

Endophytic Fungi	Fungi and Fungus-Like Taxa	Plant Association	Bioactivity	Secondary Metabolite	Class of Compound	Active Concentration	Pathogen(s)	Reference
*Pestalotiopsis foedan*	Coelomycetes	*Bruguiera* *sexangula*	Antifungal	(3R,4R,6R,7S)-7-hydroxyl-3,7-dimethyl-oxabicyclo [3.3.1] nonan-2-one	Monoterpene lactone	3.1 µg/mL (MIC)	*Botrytis cinerea* *Phytophthora nicotianae*	[65]
				(3R,4R)-3-(7-methylcyclohexenyl)-propanoic acid		6.3 µg/mL		[66]
*Pestalotiopsis* sp. DO14	Coelomycetes	*Dendrobium* *officinale*	Antifungal, Cytotoxic	(4S,6S)-6-[(1S,2R)-1,2-dihydroxybutyl]-4-hydroxy-4-methoxytetrahydro-2H-pyran-2-one	Monoterpenoid	≤25 µg/mL (MIC)	*Candida albicans* *Cryptococcus neoformans* *Trichophyton rubrum* *Aspergillus fumigatus*	[67]
				(6S,2E)-6-hydroxy-3-methoxy-5-oxodec-2-enoic acid				
*Diaporthe* *maritima*	Coelomycetes	*Picea* sp.	Antifungal	Phomopsolide A	Dihydropyrones	25 µM (MIC)	*Microbotryum violaceum*	[68]
				Phomopsolide B		250 µM		
				Phomopsolide C		250 µM		
*Scleroderma* UFSM Sc1	Basidiomycetes	*Eucalyptus* *grandis*	Antifungal, Insecticidal	Sclerodol A	Lanostane-type triterpenes	50 µg/mL (MIC) 50 µg/mL 12.5 µg/mL 25 µg/mL	*C. albicans* *C. tropicalis* *C. crusei* *C. parapsilosis*	[69]
				Sclerodol B		25 µg/mL 25 µg/mL 6.25 µg/mL 12.5 µg/mL		
*Fusarium fujikuroi* (WF5)	Hyphomycetes	*Eleusine* *coracana*	Antifungal	5-hydroxy 2(3H)-benzofuranone	Furanone	31.25 µg/mL (MIC)	*F. graminearum*	[70]
				Harpagoside	Iridoide glycoside	31.25 µg/mL		
*Trichoderma* *koningiopsis* YIM PH30002	Hyphomycetes	*Panax notoginseng*	Antifungal	Koningiopisin C	Polyketides	32 µg/mL (MIC) 64 µg/mL 32 µg/mL 16 µg/mL	*F. oxysporum* *A. panax* *F. solani* *P. cucumerina*	[71]
*Trichoderma brevicompactum* 0248	Hyphomycetes	*Allium sativum*	Antifungal	Trichodermin	Sesquiterpene	EC_50_ of 0.25 µg/mL 2.02 µg/mL 25.60 µg/mL	*R. solani* *B. cinerea* *C. lindemuthianum*	[72]
*Aspergillus* sp.	Hyphomycetes	*Gloriosa superba*	Antimicrobial, Cytotoxic	6-methyl-1,2,3-trihydroxy-7,8-cyclohepta-9,12-diene-11-one-5,6,7,8-tetralene-7-acetamide (KL-4)	Tetralene derivative	25 µg/mL (MIC) 12.5 µg/mL 50 µg/mL	*S. cerevisiae* *C. albicans* *C. gastricus*	[73]
*Penicillium* sp. R22	Hyphomycetes	*Nerium indicum*	Antifungal	3-O-methylviridicatin	Isoquinolone alkaloid	31.2 µg/mL (MIC)	*A. brassicae* *B. cinerea* *V. mali*	[74]
				Viridicatol		31.2 µg/mL	*A. brassicae* *A. alternata* *B. cinerea*	
				5-hydroxy-8-methoxy-4-phenylisoquinolin-1(2H)-one		31.2 µg/mL	*A. brassicae* *A. alternata* *V. mali*	
*Trichoderma* sp. 09	Hyphomycetes	*Myoporum* *bontioides*	Antifungal	Dichlorodiaportin	Isocoumarin	6.25–150 µg/mL(MIC)	*C. musae* *Rhizoctonia solani*	[75]
				Dichlorodiaportinolide				
*Fusarium* *chlamydosporium*	Hyphomycetes	*Anvillea garcinii*	Antimicrobial, Cytotoxic	Fusarithioamide A	Benzamide derivative	3.1 μg mL^−1^ (MIC) 4.4 μg mL^−1^ 6.9 μg mL^−1^	*B. cereus* *S. aureus* *E. coli*	[76]
*Curvularia* sp., strain M12	Hyphomycetes	*Murraya koenigii*	Antifungal	Murranofuran A	Dihydrofurans	0.5 µg/mL	*Phytophthora capsici*	[77]
				Murranolide A	Oxygenated polyketide	IC_50_ 50–100 µg/mL		
				Murranopyrone	Dihydropyrones	50–100 µg/mL		
				Murranoic acid A	Dienoic acid	50–100 µg/mL		
*Fusarium* sp.	Hyphomycetes	*Mentha* *longifolia*	Antimalarial Antifungal	Fusaripeptide A	Cyclodepsipeptide	IC_50_ 0.24 µM 0.11 µM	*C. glabrata* *C. albicans*	[78]
*Trichothecium* sp.	Hyphomycetes	*Phyllanthus* *amarus*	Anticancer, Antimetastatic, Antifungal	Trichothecinol A	Trichothecenes	20 µg/mL (MIC)	*Cryptococcus albidus* HeLa and B_16_F_10_ cells MDA-MB-_231_ cells	[79]
*Phoma* sp.	Coelomycetes	*Fucus serratus*	Antimicrobial	Phomafuranol (3R)-5-hydroxymellein Phomalacton Emodin	Dihydrofuran derivative	NR	*M. violaceum*	[80]
				(3R)-5-hydroxymellein		5 mm ZOI		
				Phomalacton		6 mm		
				Emodin		5 mm		
*Rhizopycnis vagum Nitaf* 22	Coelomycetes	*Nicotiana* *tabacum*	Antimicrobial, Cytotoxic	Rhizopycnin D	Dibenzo-α-pyrone derivatives	IC_50_ 9.9 µg/mL	*M. oryzae*	[81]
*Colletotrichum* sp.	Coelomycetes	Gomera	Antibacterial, Antifungal, Antialgal	Seimatoric acid	Oxobutanoic acid derivative	NR	*Microbotryum* v*iolaceum*	[82]
				Colletonoic acid	Benzoic acid derivative	7 mm ZOI	*B. megaterium* *C. fusca*	
*Xylaria* sp. XC-16	Ascomycetes	*Toona sinensis*	Cytotoxic, Fungicidal	Cytochalasin Z28	Cytochalasins	12.5 µM (MIC)	*G. saubinetti*	[83]
*Penicillium* *chrysogenum*	Hyphomycetes	*Cistanche* *deserticola*	Neuroprotective	Chrysogenamide A	Macfortine alkaloids	IC_50_ 1 × 10^−4^ µM	SH-SY5Y cells	[84]
				Circumdatin G				
				Benzamide				
				2′,3′-dihydrosorbicillin(9*Z*,12*Z*)-2,3-dihydroxypropyloctadeca-9,12-dienoate				
*Chaetomium* *globosum* CDW7	Ascomycetes	*Ginkgo biloba*	Antifungal	Chaetoglobosin A	Chaetoglobosins	IC_50_ 0.35 µg/mL	*S. sclerotiorum*	[85]
				Chaetoglobosin D		0.62 µg/mL		
*Coniothyrium* sp.	Coelomycetes	*Salsola* *oppostifolia*	Antimicrobial	Coniothyrinones A	Hydroxyanthraquinone	7.5 mm ZOI	*Microbotryum violaceum*	[86]
				Coniothyrinones B		6 mm		
				Coniothyrinones C		8 mm		
				Coniothyrinones D		7.5 mm		
*Pestalotiopsis fici*	Coelomycetes	*Camellia sinensis*	Antifungal	Ficipyrone A	α-pyrones	IC_50_ 15.9 µM	*Gibberella zeae*	[87]
*Xylaria* sp. strain F0010	Ascomycetes	*Abies holophylla Garcinia* *hombroniana*	Antioxidant	Griseofulvin	Indanones	IC_50_ 18.0 µg/mL	*A. mali*	[88,89]
						5.0 µg/mL	*B. cinerea*	
						1.7 µg/mL	*Colletotrichum* *gloeosporioides*	
						11.0 µg/mL	*Corticium sasaki*	
						30.0 µg/mL	*F. oxysporum*	
						1.7 µg/mL	*M. grisea*	
*Phaeoacremonium* sp.	Hyphomycetes	*Senna spectabilis*	Antifungal	Isoaigialone B Isoaigialone C	Lactone derivatives	5 µg >5 µg	*Cladosporium cladosporioides* *C. sphaerospermum*	[90]
				Aigialone		5 µg		
*Aspergillus terreus*	Hyphomycetes	*Carthamus* *lanatus*	Anti-microbial, Anti-malarial, Anti-leishmanial	(22E,24R)-stigmasta-5,7,22-trien-3-β-ol	Butyrolactones	IC_50_ 4.38 µg/mL	*C. neoformans*	[91]
				Aspernolides F		5.19 µg/mL		
*Penicillium* *raciborskii*	Hyphomycetes	*Rhododendron* *tomentosum*	Antifungal	Outovirin C	Bridged epipolythiodiket-opiperazines	0.38 µM (MIC)	*F. oxysporum* *B. cinerea* *V. dahlia*	[92]
*Mycosphaerella* sp.	Ascomycetes	*Eugenia* *bimarginata*	Antifungal	2-amino-3,4-dihydroxy-2-25-(hydroxymethyl)-14-oxo-6,12-eicosenoic acid	Eicosanoic acids	1.3 to 2.50 µg/mL (MIC)	*C. neoformans* *C. gattii*	[93]
				Myriocin		0.5 µg/mL		
*Guignardia* sp.	Ascomycetes	*Euphorbia sieboldiana*	Antifungal	Guignardone N	Meroterpenes and dioxolanone derivatives	FIC 0.23	*C. albicans*	[94]
				Guignardic acid		0.19		
*Hyalodendriella* sp.	Ascomycetes	*Populus deltoides* Marsh × *P. nigra* L.	Antimicrobial	Hyalodendriol C	Dibenzo-α-pyrones	19.22–98.47 μg/mL (MIC)	*Bacillus subtilis* *Pseudomonas lachrymans* *Ralstonia solanacearum Xanthomonas vesicatoria* *Magnaporthe oryzae*	[95,96]
				Palmariol B		16.18–92.21 μg/mL		
				TMC-264		16.24–85.46 μg/mL		
				Penicilliumolide B		17.81–86.32 μg/mL		
				Alternariol 9-methyl ether		107.19–123.19 μg/mL		
*Fusarium* sp.	Hyphomycetes	*Ficus carica*	Antifungal	Helvolic acid methyl ester	Helvolic acid derivative	12.5–25 μg/mL (MIC)	*B. cinerea* *C. gloeosporioides* *F. oxysporum f.* sp. *niveum* *Fusarium graminearum* *Phytophthora capsici*	[97]
				Helvolic acid				
				Hydrohelvolic acid				
*Lopherdermium nitens* DAOM 250027	Ascomycetes	*Pinus strobus*	Antifungal	Six phenolic bisabolane-type sesquiterpenoids	Phenolic bisabolane-type sesquiterpenoids	50 μM (MIC)	*Microbotryum violaceum*	[98]
				Pyrenophorin	Macrolide	5 μM	*Saccharomyces cerevisiae*	
*Epicoccum* sp.	Ascomycetes	*Theobroma cacao*	Antimicrobial, Antifungal	Epicolactone	Polyoxygenated polyketides	20–80 μg per paper disc (MIC)	*Pythium ultimum* *Aphanomyces cochlioides* *Rhizoctonia solani*	[99]
				Epicoccolide A				
				Epicoccolide B				
*Botryosphaeria dothidea* KJ-1	Ascomycetes	*Melia azedarach*	Antifungal, Antibacterial, Antioxidant Cytotoxic	Stemphyperylenol	α-pyridone derivative Ceramide derivative	1.57 μM (MIC)	*Alternaria solani*	[100]
				Pycnophorin		6.25–25 μM		
				Chaetoglobosin C				
				Djalonensone				
				Alternariol				
				β-sitosterol glucoside				
				5-hydroxymethylfurfural				
*Botryosphaeria* sp. P483	Ascomycetes	*Huperzia serrata*	Antifungal, Nematicidal	Botryosphaerin H	Tetranorlabdane diterpenoids	ZOI 9, 7, 7, 8, 8 mm	*Gaeumannomyces graminis* *Fusarium solani* *Pyricularia oryzae Fusarium moniliforme* *F. oxysporum*	[101]
				13,14,15,16-tetranorlabd-7-en-19,6β:12,17-diolide		12, 10, 10, 11, 13 mm		
*Colletotrichum gloeosporioides*	Coelomycetes	*Michelia* *champaca*	Antifungal	2-phenylethyl 1H-indol-3-yl-acetate	----	5 µg 25 µg (MIC)	*Cladosporium cladosporioides* *C. sphaerospermum*	[102]
				Uracil				
				Cyclo-(S*-Pro-S*-Tyr) Cyclo-(S*-Pro-S*-Val) 2(2-aminophenyl) acetic acid				
				4-hydroxy-benzamide				
				2(2-hydroxyphenyl) acetic acid				
*Phoma* sp. WF4	Coelomycetes	*Eleusine* *coracana*	Antifungal	Viridicatol	Viridicatol alkaloid	ZOI 1.8 mm	*Fusarium graminearum*	[103]
				Tenuazonic acid	Tenuazonic acid	2 mm		
				Alternariol	Alternariol	1.5 mm		
				Alternariol monomethyl ether	Ether derivative	1.5 mm		
*Chaetomium cupreum* ZJWCF079	Ascomycetes	*Macleaya cordata*	Antifungal	Ergosta-5,7,22-trien-3beta-ol	NR	EC_50_ 125 µg/mL 190 µg/mL	*Sclerotinia sclerotiorum* *B. cinerea*	[104]
*Microsphaeropsis* sp.	Coelomycetes	*Salsola* *oppositifolia*	Antifungal	Microsphaerol	Polychlorinated triphenyl diether	ZOI 9 and 5 mm 9 and 7 mm 8 and 3 mm	*Microbotryum violaceum* *B. megaterium* *E. coli*	[105]
*Seimatosporium* sp.				Seimatorone	Naphthalene derivative			
*Mycosphaerella* sp.	Ascomycetes	*Eugenia* *bimarginata* DC.	Antifungal	2-amino-3,4-dihydroxy-2-25-(hydroxymethyl)-14-oxo-6,12-eicosenoic acid	Eicosanoic acid	1.3–2.50 µg/mL (MIC)	*C. neoformans* *C. gattii*	[93]
				Myriocin		0.5 µg/mL		
*Pezicula* sp.	Ascomycetes	*Forsythia* *viridissima*	Antifungal	Mellein	NR	EC_50_ 48.63 µg/mL	*B. cinerea*	[106]
						150.90 µg/mL	*Colletotrichum orbiculare*	
						163.37 µg/mL	*Verticillium dahliae*	
						159.09 µg/mL	*Fusarium oxysporium* f. sp.	
						118.83 µg/mL	*Cucumerinum*	
						161.04 µg/mL	*Pyricularia oryzae*	
						125.36 µg/mL	*Pestalotia diospyri*	
						205.01 µg/mL	*Pythium ultimum*	
						45.98 µg/mL	*Sclerotinia sclerotiorum*	
*Nodulisporium* sp. A21	Ascomycetes	*Ginkgo biloba*	Anti-phytopathogenic, Antifungal	Sporothriolide	NR	EC_50_ 3.04 µg/mL 200 µg/mL	*Rhizoctonia solani* *Magnaporthe oryzae*	[107]
*Echinacea purpurea*	Ascomycetes	*Biscogniauxia mediterranea* EPU38CA	Antifungal	(−)-5-methylmellein	NR	300 µM (MIC)	*P. obscurans*	[108]
						----	*P*. *viticola*	
				(−)-(3R)-8-hydroxy-6-methoxy-3,5-dimethyl-3,4-dihydroisocoumarin	Coumarin	----	*B. cinerea*	
						300µM	*P. viticola*	
						----	*P. obscurans*	
*Phialophora mustea*	Ascomycetes	*Crocus sativus*	Antimicrobial, Cytotoxic	Phialomustin C	Azaphilone derivative	IC_50_ 14.3 µM	*Candida albicans*	[109]
				Phialomustin D		73.6 µM		
*Plectophomella* sp. *Physalospora* sp.	Ascomycetes	NR	Antifungal, Antibacterial, Herbicidal	(−)-Mycorrhizin A	Mycorrhizin	---	*Ustilago violacea* *Eurotium repens*	[110]
				Cytochalasins E	Cytochalasins	---	*E. repens* *Mycotypha microspore*	
				Cytochalasins K				
				Radicinin	Dihydropyranone	---	*E. repens* *M. microspore*	
*Berkleasmium* sp.	Ascomycetes	*Dioscorea* *zingiberensis*	Antifungal	Diepoxin ζ	Spirobisnaphthalenes	IC_50_ 9.1–124.5 µg/mL	*M. oryzae*	[111]
				Palmarumycin C11				
				Palmarumycin C12				
				Cladospirone B				
				Palmarumycin C6				
				1,4,7β-trihydroxy-8-(spirodioxy-1′,8′-naphthyl)-7,8-dihydronaphthalene				
				Palmarumycin C8				
*Diaporthe melonis*	Ascomycetes	*Annona* *squamosa*	Antimicrobial	Diaporthemins A	Dihydroanthracenone atropodiastereomers	NA	*S. aureus* 25697 *S. aureus* ATCC 29213 *S. pneumoniae* ATCC 49619	[112]
				Diaporthemins B		NA	*S. aureus* 25697 *S. aureus* ATCC 29213 *S. pneumoniae* ATCC 49619	
				Flavomannin-6,6′-di-O-methyl ether		32 μg/mL (MIC)	*Staphylococcus aureus* 25697	
						32 μg/mL	*S. aureus* ATCC 29213	
						2 μg/mL	*Streptococcus pneumoniae* ATCC 49619	
*Cryptosporiopsis quercina*	Ascomycetes	*Tripterigium wilfordii*	Antifungal	Cryptocandin	Lipopeptide	MIC 0.03–0.07 μg/mL	*C. albicans,* *Trichophyton mentagrophytes,* *Trichophyton rubrum*	[113]
*Colletotrichum gloeosporioides*	Ascomycetes	*Artemesia* *mongolica*	Antifungal	Colletotric acid	Benzoic acid derivative	MIC 25 μg/mL	*Bacillus subtilis*	[114]
						50 μg/mL	*Staphylococcus aureus*	
						50 μg/mL	*Sarcina lutea*	
						50 μg/mL	*Helminthosporium sativum*	
*Alternaria* sp.	Ascomycetes	*Trixis vauthieri*	Anti-trypanosomiasis, Anti-leishmaniasis	Altenusin	Biphenyl fungal metabolite	IC_50_ 4.3 μM	Trypanothione reductase (TR) inhibitory activity	[115]
*Aspergillus* sp. strain CY725	Hyphomycetes	NR	Antibacterial	Helvolic acid	Helvolic acid	8.0 µg/mL (MIC)	*H. pylori*	[116]
				Monomethylsulochrin		10.0 µg/mL		
				Ergosterol	Sterols	20.0 µg/mL		
				3b-hydroxy-5a, 8a-epidioxy-ergosta-6, 22-diene		30.0 µg/mL		
*Fusarium* sp. IFB-121	Hyphomycetes	*Quercus* *variabilis*	Antibacterial	Fusaruside	Cerebrosides	3.9 µg/mL (MIC)	*B. subtilis*	[117]
						3.9 µg/mL	*E. coli*	
						1.9 µg/mL	*P. fluorescens*	
				(2S,2′R,3R,3′E,4E,8E)-1-O-beta-D-glucopyranosyl-2-N-(2′-hydroxy-3′-octadecenoyl)-3-hydroxy-9-methyl-4,8-sphingadienine		7.8 µg/mL	*B. subtilis*	
						3.9 µg/mL	*E. coli*	
						7.8 µg/mL	*P. fluorescens*	
*Periconia* sp.	Ascomycetes	*Taxus cuspidata*	Antibacterial	Periconicins A	Fusicoccane diterpenes	3.12 µg/mL (MIC)	*Klebsiella pneumoniae*	[118]
				Periconicins B		25 µg/mL		
*F. oxysporum*	Hyphomycetes	*Lycopersicum* *esculentum*	Nematicidal	3-hydroxypropionic acid	Propionic acid	LD_50_ 12.5–15 µg/mL	*M. incognita*	[119]
*Chaetomium* sp.	Ascomycetes	*Nerium oleander*	Antioxidant	NR	NR	IC_50_ 109.8 μg/mL	Inhibited xanthine oxidase activity	[120]
*Cladosporium cladosporioides*	Ascomycetes	*Huperzia serrata*	Prevent neurodegeneration	Huperzine A	NR	10 μg/mL	Acetylcholinesterase inhibition activity	[121]
*Edenia* sp.	Ascomycetes	*Petrea volubilis*	Antiparasitic	Palmarumycin CP17	NR	IC_50_ 1.34 μM	*Leishmania donovani*	[122]
				Palmarumycin CP18		0.62 μM		
*Nodulisporium* sp.	Ascomycetes	*Erica arborea*	Antifungal, Antialgal	Nodulisporins D	Naphthalene-Chroman Coupling products	ZOI 8 mm	*B. megatarium*	[123]
						7 mm	*M. violaceum*	
						8 mm	*C. fusca*	
				Nodulisporins E		7 mm	*B. megatarium*	
						7 mm	*M. violaceum*	
						5 mm	*C. fusca*	
				Nodulisporins F		8 mm	*B. megatarium*	
						10 mm	*M. violaceum*	
						8 mm	*C. fusca*	
				(3S,4S,5R)-2,4,6-trimethyloct-6-ene-3,5-diol		0 mm	*B. megatarium*	
						8 mm	*M. violaceum*	
						6 mm	*C. fusca*	
				5-hydroxy-2-hydroxymethyl-4H-chromen-4-one		0 mm	*B. megatarium*	
						6 mm	*M. violaceum*	
						6 mm	*C. fusca*	
				3-(2,3-dihydroxyphenoxy)-butanoic acid		0 mm	*B. megatarium*	
						6 mm	*M. violaceum*	
						7 mm	*C. fusca*	
*Chalara* sp.	Ascomycetes	*Artemisia* *vulgaris*	Antibacterial	Isofusidienol A	Chromone-3-oxepines	ZOI 23 mm	*B. subtilis*	[124]
				Isofusidienol B		22 mm		
				Isofusidienol C		9 mm		
				Isofusidienol D		8 mm		
*Ampelomyces* sp.	Ascomycetes	*Urospermum picroides*	Cytotoxic	6-O-methylalaternin	NR	41.7 μM (MIC)	*S. aureus* *S. epidermidis* *Enterococcus faecalis*	[125]
				Altersolanol A		37.2–74.4 μM		
*Alternaria* sp.	Ascomycetes	*Sonneratia alba*	Antibiotic	Xanalteric acid I	Phenolic compounds	343.40–686.81 μM	*S. aureus*	[126]
				Xanalteric acid II				
*Pestalotiopsis theae*	Coelomycetes	---	Anti-HIV	Pestalotheol A	NR	NR	HIV-1LAI replication in C8166 cells	[127]
				Pestalotheol B		NR		
				Pestalotheol C		EC_50_ 16.1 μM		
				Pestalotheol D		NR		
*Stemphylium* *globuliferum*	Ascomycetes	*Mentha pulegium*	Cytotoxic	Alterporriol G and H (mixture)	NR	EC_50_ 2.7 μM	L5178Y lymphoma cells	[128]
				Altersolanol K				
				Altersolanol L				
				Stemphypyrone				
*Chaetomium* sp.	Ascomycetes	*Otanthus* *maritimus*	Cytotoxic	Aureonitolic acid	Tetrahydrofuran derivative	NA	L5178Y mouse lymphoma cells	[129]
				Cochliodinol		EC_50_ 7.0 μg/mL		
				Isocochliodinol		---		
				Iindole-3-carboxylic acid		---		
				Cyclo(alanyltryptophane)		---		
				Orsellinic acid		2.7 μg/mL		
*Chaetomium* sp.	Ascomycetes	*Eucalyptus* *exserta*	Antibacterial	Mollicellin O	Depsidones	IC_50_ 79.44 μg/mL	*S. aureus* ATCC29213	[130]
						76.35 μg/mL	*S. aureus* N50	
				Mollicellin H		5.14 μg/mL	*S. aureus* ATCC29213	
						6.21 μg/mL	*S. aureus* N50	
				Mollicellin I		70.14 μg/mL	*S. aureus* ATCC29213	
						63.15 μg/mL	*S. aureus* N50	
			Cytotoxic	Mollicellin G		19.64 μg/mL	HepG2 cell line	
						13.97 μg/mL	Hela cell line	
				Mollicellin H		6.83 μg/mL	HepG2 cell line	
						---	Hela cell line	
			Antioxidant	Mollicellin I		---	HepG2 cell line	
						21.35 μg/mL	Hela cell line	
				Mollicellin O		71.92 μg/mL	DPPH free radical scavenging assay	
*Alternaria* sp.	Ascomycetes	*Polygonum* *senegalense*	Cytotoxic	Alternariol	Sulfated derivatives of alternariol and its monomethyl ethers	EC_50_ 1.7 μg/mL	L5178Y lymphoma cells	[131]
				Alternariol 5-O-sulfate		4.5 μg/mL		
				Alternariol 5-O-methyl ether Altenusin Desmethylaltenusin		7.8 μg/mL		
				2,5dimethyl-7-hydroxychromone Tenuazonic acid		6.8 μg/mL		
				Altertoxin I		6.2 μg/mL		
				3′-hydroxyalternariol 5-O-methyl ether		---		
				Alterlactone		---		
				Alternaric acid		---		
				Talaroflavone		---		
				Altenuene		---		
				4′-epialtenuene		---		
*Phyllosticta* *spinarum*	Ascomycetes	*Platycladus* *orientalis*	Cytotoxic	Tauranin(+)-(5 S,10 S)-4′-hydroxymethylcyclozonarone	Sesquiterpene quinones	EC_50_ 4.3 μM	NCI-H460 cell lines MCF-7 cell lines SF-268 cell lines PC-3 M cell lines MIA Pa Ca-2 cancer cell lines	[132]
				3-ketotauranin		1.5 μM		
				3alpha-hydroxytauranin		1.8 μM		
				12-hydroxytauranin		3.5 μM		
				Phyllospinarone		2.8 μM		
*Eupenicillium* sp.	Ascomycetes	*Glochidion* *ferdinandi*	Cytotoxic	Trichodermamide C	A modified dipeptide	IC_50_ 1.5 μM	Human colorectal carcinoma cell line HCT116	[133]
						9.3 μM	Human lung carcinoma cell line A549	
*Pestalotiopsis* sp.	Coelomycetes	*Rhizophora* *mucronata*	Cytotoxic	Pestalotiopsone F	Pestalasins chromones	EC_50_ 26.89 μM	Murine cancer cell line L5178Y	[134,135]

Abbreviations: ZOI, Zone of Inhibition; MIC, Minimum Inhibitory Concentration; IC_50_, Inhibitory Concentration T 50%; EC_50_, Effective Concentration at 50%; LD_50_, Lethal Dose Concentration at 50%; FIC, Fractional Inhibitory Concentration; NR, Not Reported; NA, Not Active.

**Table 3 microorganisms-10-00360-t003:** Commercially available drugs from Endophytic fungi: key examples, pharmacological function, bottlenecks, and success stories.

Marketed Drug			Commerical Market	Endophyte	Pharmacological Functions	Bottlenecks	References
Taxol (Paclitaxel) 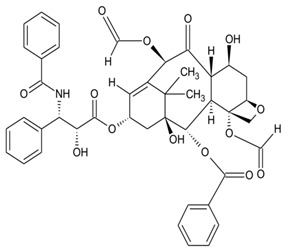			$78.77 million (2017)	*Taxomyces andreanae* *Taxus brevifolia*	Anticancer-binds to microtubule assembly and delays cell division/growth, Treatment for breast, ovary, and Kaposi’s sarcoma	Biosynthetic complexities, harvestation of the molecule	[15,194,195]
Camptothecin 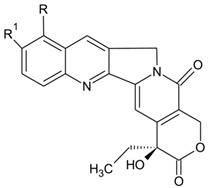			$2.2 billion (2008)	*Entrophospora infrequens* *Alternaria alternata* *Fomitopsis* sp. *Phomopsis* sp.	Inhibitor of Topoisomerase enzyme	Yield-loss on repeated sub-culturing	[196,197]
	R	R1					
Camptothecin (2)	H	H					
9 Methoxycamptothecin (3)	OCH3	H					
10-Hydroxycamptothecin (4)	H	OH					
Vinca alkaloids (Vinblastine and Vincristine) 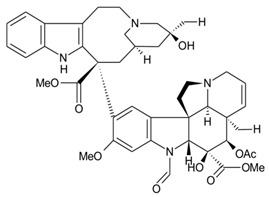			-	*Alternaria* sp. *F. oxysporum*	Anticancer alkaloids, inhibit microtubule assembly leading to destabilization	Yield-loss on repeated sub-culturing	[198,199]
Podophyllotoxin 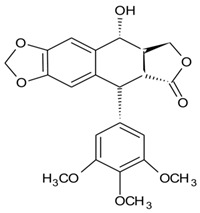			$444.1 million (2019)	*Phialocephala fortinii*	Cytotoxic in U-87 cell line, Antitumor activity in cancer models	Low abundance in plants	[200,45]
Griseofulvin 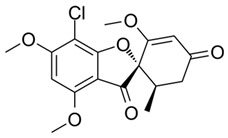			$5.8 billion (2019)	*Xylaria* sp. strain F0010	Antioxidant	Low abundance, production from alternate sources	[89]
Laritin A and B 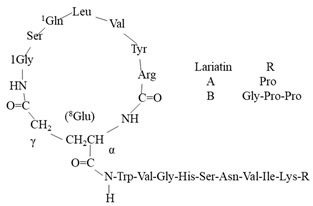			-	*Rhodococcus jostii*	Anti-inflammatory, Anticancer, Immunomodulatory properties	Low abundance, Yield-loss on repeated sub-culturing	[13]
Rohitukine 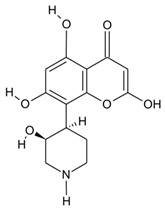			-	*Fusarium proliferatum*	Cytotoxic against the HCT-116 and MCF7 cell lines	Yield-loss on repeated sub-culturing	[201]
Piperine 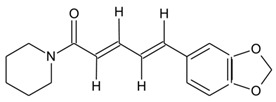			$4.87 billion (2020)	*Periconia* strains	Hepatoprotective Antibacterial	Yield-loss on repeated sub-culturing	[202] https://www.expertmarketresearch.com/reports/piperine-market accessed on 12 July 2021
Altersolanol 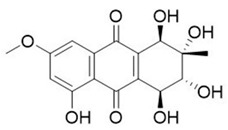			-	*Alternaria* sp.	Antiangiogenic	Yield-loss on repeated sub-culturing	[203]
Huperzine A 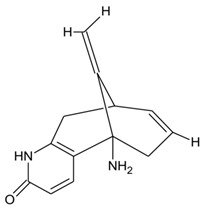			-	*Shiraia* sp.	Cholinesterase inhibitor	Yield-loss on repeated sub-culturing	[204]
Phomoxanthone A and B 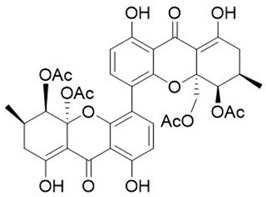			-	*Phomopsis* sp., BCC 1323	Cytotoxic against BC-1 cells, KB, and Vero (non-malignant) cells	Yield-loss on repeated sub-culturing	[205]
Quinine 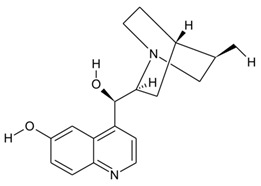			$804.98 million	*Phomopsis* sp.	Antimalarial agent	Yield-loss on repeated sub-culturing	[206] https://www.prnewswire.com/news-releases/global-quinine-market-review-2015-2018 accessed on 12 July 2021
Silybin A 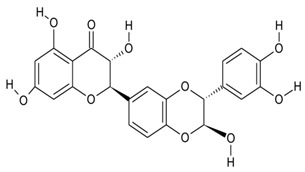			-	*Aspergillus iizukae*	Hepatoprotective	Low abundance, production from alternate sources	[207]

## Data Availability

Not applicable.

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
