# Peer review of "Endophytic Fungi: Key Insights, Emerging Prospects, and Challenges in Natural Product Drug Discovery"

_microorganisms, 2022, doi:10.3390/microorganisms10020360_

Round 1
Reviewer 1 Report
Dear authors and editors,
Here is the review of the paper titled "Endophytic Fungi: Key insights, Emerging prospects, and Challenges in Natural Product Drug Discovery" written by Pragya Tiwari & Hanhong Bae.
It is a review paper dealing with endophytic fungi that live inside plants and produce a vast array of biologically active secondary metabolites used in diverse industries for benefit of humankind. It is a very thorough review definitely worthwhile publishing in Microorganisms journal. But it needs a major revision.
The most important points that need to be corrected are:
(1) A few parts of the text discuss some taxa from the genus Pseudomonas. Since this genus belongs to Bacteria (Prokaryotes) and the paper mentions only fungi (Eukaryotes) in the title the authors should omit these parts of the text from the manuscript.
(2) The tables are pretty much messed up and should be thoroughly rechecked and much better organized to allow readers to easily follow the text. While reading the tables, the text belonging to one reference cited is sometimes dispersed in a few rows, thus I am not sure which reference refers to which Biological application, Endophytic fungi, Plant species, and Outcome (e.g. manuscript page 4, Table 1.). A similar situation is with Table 2.
The English language is pretty well.
My other minor suggestions/corrections are included in the manuscript pdf.
The paper needs major revision.
Best, Reviewer

Reviewer 2 Report
The manuscript titled "Endophytic Fungi: Key insights, Emerging prospects, and Challenges in Natural Product Drug Discovery" covers an interesting group of microorganisms producing bioactive metabolites with pharmacological applications. The authors cited many references in the field but I do not see other reviews in this field and it would be nice if they mention previous reviews on this topic and how this review differs from them. Also, the authors mentioned outdated statistics about the number of natural products from endophytes. Therefore, the manuscript needs to be revised before acceptance. Below are some additional comments which need to be addressed.
Line 40: This statistics looks old for this review. The authors should provide recent statistics.
Line 56: This Figure is not clear in this shape. The grey and orange gear wheels are non-sense after Bioprospection. Instead showing the origin and usage of the metabolites would be more important.
Line 76: It is important to add the strain of the fungal species in Table 1.
Line 108-109: However,…is this sentence about phytochemicals or all bioactive metabolites? As mentioned only plant species!
Line 138: “Endophytic fungi produce diverse kinds of bioactive metabolites; however, there is little change in their biosynthetic pathways via a few precursor compounds and are categorized as terpenoids, polyketides, and non-ribosomal peptides”. This sentence is not clear! Probably authors wanted to say …“Endophytic fungi diverse kinds of bioactive metabolites most notably terpenoids, polyketides, and non-ribosomal peptides”. I do not understand “however, there is little change in their biosynthetic pathways via a few precursor compounds”.
Line 140-144: First of all: The mevalonate pathway only provides the substrates for different terpene synthases/cyclases to produce different terpenoids. Second, this is independent from Polyketide synthase in terms of secondary metabolism. Third, Non-ribosomal peptide synthetases use different amino acids to form the non-ribosomal peptides. Additionally, how did the authors claim other classes of metabolites and analogs are biosynthesized from the above classes? There are many independent classes of compounds! Authors must reword this paragraph in a scientific way.
Line 150: again these statistics are old.
Line 169: Please mention these metabolites from 2010-2017 which shows the recent discoveries in this field.
Line 207: Are there no polyketides or non-ribosomal peptides isolated from endophytic fungi? Authors mentioned them before as main bioactive metabolites together with terpenoids!
Line 244: This figure misses the bioengineering methodologies mentioned before!
Line 249: Deep learning! Here is for traditional methods!
Line 344: Draw the structures with the same style! Some structures are simply copied to the table as pictures or differ in style.
Line 351: 2.5 g not 2.5 gm!
Round 2
Reviewer 1 Report
Dear authors & editor,
The paper is very much revised according to my suggestions. However, there are a few errors still present in the text:
- Table 1. Trichoderma peudokoningi -> Trichoderma pseudokoningii
- Table 1. Mangrove sp. - Mangrove is not a plant genus, change to Mangrove vegetation or similar phrase
- Table 1. "T. brevifolia" should be deleted in column endophytic fungi while in column plant species: Pacific Yew -> Taxus brevifolia
Other parts of the text and tables seem fine with me. The paper can be accepted after the minor revision.
Best, reviewer
Reviewer 2 Report
The review manuscript titled " Endophytic Fungi: Key insights, Emerging prospects, and Challenges in Natural Product Drug Discovery" has been improved significantly and I have no more comments on it.